# Discrete Denoising Flows

**Alexandra Lindt** [1]   **Emiel Hoogeboom** [1]

## Abstract

Discrete flow-based models are a recently proposed class of generative models that learn invertible transformations for discrete random variables. Since they do not require data dequantization and maximize an exact likelihood objective, they can be used in a straight-forward manner for lossless compression. In this paper, we introduce a new discrete flow-based model for categorical random variables: *Discrete Denoising Flows* (DDFs). In contrast with other discrete flow-based models, our model can be locally trained without introducing gradient bias. We show that DDFs outperform Discrete Flows on modeling a toy example, binary MNIST and Cityscapes segmentation maps, measured in log-likelihood.

## 1. Introduction

Due to their wide range of applications, flow-based generative models have been extensively studied in recent years (Rezende & Mohamed, 2015; Dinh et al., 2016). Research has mainly focused on modeling continuous data distributions, in which discretely stored data like audio or image data must be dequantized prior to modeling. However, two recent publications explore flow-based generative modeling of discrete distributions: *Discrete Flows* (Tran et al., 2019) for categorical random variables and *Integer Discrete Flows* (Hoogeboom et al., 2019) for ordinal discrete random variables. Due to their discrete nature and exact likelihood objective, these discrete flow-based models can be used directly for lossless compression.

Unlike other approaches that use generative models for lossless compression, discrete flow-based models are advantageous because they (i) enable efficient inference and (ii) can encode single data samples efficiently. Approaches that use

---

[1]UvA-Bosch Delta Lab, University of Amsterdam, Amsterdam Netherlands. Correspondence to: Alexandra Lindt <alex.lindt@protonmail.com>, Emiel Hoogeboom <e.hoogeboom@uva.nl>.

Third workshop on *Invertible Neural Networks, Normalizing Flows, and Explicit Likelihood Models* (ICML 2021). Copyright 2021 by the author(s).

the *Variational Autoencoder* (VAE) (Kingma & Welling, 2013) for lossless compression typically combine the model with bits-back-coding (Hinton & Van Camp, 1993), which is effective for compressing full data sets but inefficient for encoding single samples. Autoregressive models such as PixelCNN (Oord et al., 2016) can also be used for lossless compression, however, they are generally expensive to decode.

Unfortunately, both *Discrete Flows* and *Integer Discrete Flows* come with the drawback that each of their layers contains a quantization operation. When optimizing them with the backpropagation algorithm, the gradient of the quantization operation has to be estimated with a biased gradient estimator, which may compromise their performance.

To improve training efficiency, reduce gradient bias and improve overall performance, we introduce a new discrete flow-based generative model for categorical random variables, *Discrete Denoising Flows* (DDFs). DDFs can be trained without introducing gradient bias. They further come with the positive side effect that the training is computationally very efficient. This efficiency results from the local training algorithm of the DDFs, which trains only one layer at a time instead of all at once. We demonstrate that *Discrete Denoising Flows* outperform *Discrete Flows* in terms of log-likelihood.

## 2. Related Work & Background

This section first introduces normalizing flows as well as discrete flows. It then goes on to describe alternate approaches that use generative models for lossless compression.

**Normalizing Flows**   The fundamental idea of flow-based modeling is to express a complicated probability distribution as a transformation on a simple probability distribution. Given the two continuous random variables $X$ and $Z$ and the invertible and differentiable transformation $T : \mathcal{Z} \rightarrow \mathcal{X}$, $X$'s probability distribution $p_X(\cdot)$ can be written in terms of $Z$'s probability distribution $p_Z(\cdot)$ as

$$p_X(x) = p_Z(z) \left| \det J_T(z) \right|^{-1} \quad \text{with } z = T^{-1}(x), \quad (1)$$

using the change of variables formula. The Jacobian determinant acts as *normalizing* term and ensures that $p_X(\cdot)$ is a valid probability distribution. The distribution $p_Z(\cdot)$ is

referred to as the base distribution and the transformation $T$ as a *normalizing flow*. A composition of invertible and differentiable functions can be viewed as a repeated application of formula 1. Therefore, such compositions are also referred to as *normalizing flows* throughout literature.

**Discrete Flows**  In the case of two discrete random variables $X$ and $Z$, the change of variables formula for continuous random variables given in formula 1 simplifies to

$$p_X(x) = p_Z(z) \quad \text{with} \quad z = T^{-1}(x) \quad (2)$$

Normalization with the Jacobian determinant is no longer necessary as it corrects for a change in volume. Discrete distributions, however, have no volume since they only have support on a discrete set of points.

*Discrete Flows* (DFs) (Tran et al., 2019) are discrete flow-based models that learn transformations on categorical random variables. The authors define a bijective transformation $T : \mathcal{Z} \rightarrow \mathcal{X}$ with $\mathcal{X} = \mathcal{Z} = \{1, \ldots, K\}^D$ in the form of a bipartite coupling layer (Dinh et al., 2016). The coupling layer input $x$ is partitioned into two sets s.t. $x = [x_a, x_b]$ and then transformed into an output $z = [z_a, z_b]$ with

$$\begin{aligned} z_a &= x_a \\ z_b &= (s_{\theta_1}(x_a) \circ x_b + t_{\theta_2}(x_a)) \bmod K \end{aligned} \quad , \quad (3)$$

where $\circ$ denotes element-wise multiplication. Note that the transformation is only invertible if each element of the scale is coprime to the number of classes $K$. Scale $s_{\theta_1}(\cdot)$ and translation $t_{\theta_2}(\cdot)$ are modeled by a neural network with parameters $\theta_{1,2}$. To obtain discrete scale and translation values, the authors use the argmax operator combined with a relaxed softmax as gradient estimator (Jang et al., 2016) to enable backpropagation. This introduces bias to the model parameter gradients, which harms optimization. Note that the example describes the bipartite version and not the autoregressive version of DFs. In (Tomczak, 2020) different partitions for coupling layers are explored.

**Generative Models for Lossless Compression**  The *Variational Autoencoder* (VAE) (Kingma & Welling, 2013) can be used for lossless compression is by discretizing the continuous latent vector and applying bits-back coding (Hinton & Van Camp, 1993). Recent methods that work according to this approach include Bits-Back with ANS (Townsend et al., 2019a), Bit-Swap (Kingma et al., 2019) and HiLLoC (Townsend et al., 2019b). These methods obtain performances close to the negative ELBO for compressing full datasets. However, when encoding a single data sample they are rather inefficient because the auxiliary bits needed for bits-back coding cannot be amortized across many samples. The same problem but in a scaled-up version due to multiple latent variables arises when local bits-back coding is used in normalizing flows (Ho et al., 2019). In this case, encoding

a single image would require more bits than the original image. Mentzer et al. (Mentzer et al., 2019) use a VAE with deterministic encoder to transform a data sample into a set of discrete multiscale latent vectors. Although this method does not require bits-back coding, it optimizes only a lowerbound on the likelihood instead of the likelihood directly. Another generative model that is well suited for lossless compression is the PixelCNN (Oord et al., 2016). PixelCNN organizes the pixels of an image as a sequence and predicts the distribution of a pixel conditioned on all previous pixels. Consequently, drawing samples from PixelCNN requires multiple network evaluations and is very costly. Nevertheless, PixelCNN achieves state-of-the-art performances in lossless compression.

## 3. Method: Discrete Denoising Flows

In this section, we introduce *Discrete Denoising Flows*. Like other flow-based models, DDFs consist of several bipartite coupling layers that are easily invertible, the so-called *denoising coupling layers*.

### 3.1. Denoising Coupling Layer

Complying to the change of variables formula 2, we define the denoising coupling layer as an invertible transformation $T : \mathcal{Z} \rightarrow \mathcal{X}$ between two categorical variables $X$ and $Z$ with domains $\mathcal{X} = \mathcal{Z} = \{1, \ldots, K\}^D$. The inverse $T^{-1}$, representing the forward pass during training, is given as

$$\begin{aligned} z_a &= x_a \\ z_b &= \text{cond\_perm}(x_b | n(x_a)) \end{aligned} \quad (4)$$

That is, the input $x \in \{1, .., K\}^D$ is partitioned into two sets such that $x = [x_a, x_b]$ and $x_a \in \{1, .., K\}^d$. The first part stays the same while the second part is transformed conditioned on the first part. For this transformation, we use a neural network $n$ as well as the conditional permutation operation $\text{cond\_perm}(\cdot | \cdot)$.

The conditional permutation operation is the core component of the denoising coupling layer. For notation clarity, we define the variable $\theta = n(x_a)$ with $\theta \in \mathbb{R}^{(D-d) \times K}$ as the output of the neural network $n$. The conditional permutation operation is then defined as

$$\text{cond\_perm}_i(x_{b_i} | \theta_i) = \text{perm}_{\theta_i}(x_{b_i}) \quad (5)$$

$$\text{perm}_{\theta_i} = \begin{pmatrix} 1 & 2 & \ldots & K \\ \text{argsort\_top}_h(\theta_i) \end{pmatrix} \quad (6)$$

per dimension $i \in \{1, \ldots, D-d\}$, where we used Cauchy's two-line notation for permutations to define the permutation $\text{perm}_{\theta_i}$. We also introduced the $\text{argsort\_top}_h(\cdot)$ operation with the additional hyperparameter $h \in \{1, .., K\}$. This operation acts similar to the regular argsort operation, with

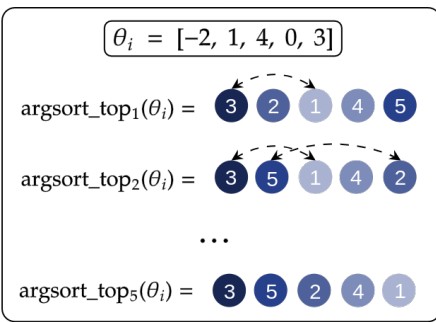

*Figure 1.* Functionality of $\text{argsort\_top}_h(\theta_i)$ illustrated for an example $\theta_i$ with number of classes $K = 5$.

the difference that it only sorts the top $h$ largest elements while leaving the remaining elements in their previous positions. Figure 1 illustrates this functionality. The intuition behind the operation is that only the most predictable classes are permuted, leaving more of the structure intact than an entire argsort. Also, observe that in the binary case $K = 2$ and for $h = K$, argsort and $\text{argsort\_top}_h$ are equivalent.

The conditional permutation operation is easily invertible as

$$\text{cond\_perm}_i^{-1}(x_{b_i}|\theta_i) = \text{perm}_{\theta_i}^{-1}(x_{b_i}) \qquad (7)$$

$$\text{perm}_{\theta_i}^{-1} = \begin{pmatrix} \text{argsort\_top}_h(\theta_i) \\ 1 \quad 2 \quad \ldots \quad K \end{pmatrix} \qquad (8)$$

per dimension $i \in \{1, \ldots, D-d\}$. Using this definition, we can write the transformation $T$ representing the denoising coupling layer at inference time as

$$\begin{aligned} x_a &= z_a \\ x_b &= \text{cond\_perm}^{-1}(z_b|n(z_a)) \end{aligned} \qquad (9)$$

### 3.2. Training Denoising Discrete Flows

For training a denoising coupling layer, we simply train a neural network $n$ to predict $x_b$ from $x_a$. To this end, we use the mean cross-entropy loss between $n(x_a)$ and $x_b$ as our objective function. After training, the fixed neural network $n$ can be employed in a denoising coupling layer. When we apply the conditional permutation operation in

$$z_b = \text{cond\_perm}(x_b|n(x_a)),$$

the more the argmax of $n(x_a)$ resembles $x_b$, the more likely it is for a value in $x_b$ to be switched to one of the smaller class values. Consequently, given that the argmax of $n(x_a)$ somewhat resembles $x_b$, the outcome of the conditional permutation operation $z_b$, is more likely to contain smaller class values than $x_b$. This makes the value of the random variable $Z$ more predictable than the value of the random variable $X$, when looking at those dimensions in isolation. In other words, we decorrelated the random variable $X$ into the random variable $Z$. As a direct consequence, modeling the

**Algorithm 1** transform(ddf, $S$, $x$). Transforms $x \mapsto z$

**Input:** $x$, ddf     // ddf is a list of classifiers $[n_1, n_2, \ldots]$
Let $z = x$
**for** $n$, shuffle in ddf, $S$ **do**
    Split $[z_a, z_b] = z$
    $z_b = \text{cond\_perm}(z_b|\theta = n(z_a))$
    Combine $z = [z_a, z_b]$
    $z = \text{shuffle}(z)$
**end for**
**return** $z$

---

**Algorithm 2** optimize($n_{\text{new}}$, $z$). Optimize a new layer.

**Input:** $n_{\text{new}}$, $z$     // $n_{\text{new}}$ is a pixel-wise classifier
Split $[z_a, z_b] = z$
Optim. $\log \mathcal{C}(z_b|\theta = n_{\text{new}}(z_a))$     // Equiv. to cross-entropy

---

**Algorithm 3** Training DDFs

**Input:** number of layers $L$
ddf $= [\,]$     // create DFF with 0 layers
Init $S = [\text{shuffle}_1, \ldots \text{shuffle}_L]$     // Init $L$ shuffling layers
**while** $i < L$ **do**
    Init $n_{\text{new}}$     // new classifier
    **while** $n_{\text{new}}$ not converged **do**
        Sample $x \sim$ Data
        $z = \text{transform}(\text{ddf}, S, x)$
        optimize($n_{\text{new}}$, $z$)
    **end while**
    ddf.append($n_{\text{new}}$)
**end while**
**return** ddf

---

distribution $p_Z(\cdot)$ with a $D$-dimensional i.i.d. categorical distribution will result in a smaller mismatch than it would for the distribution $p_X(\cdot)$. To give some more intuition on this functionality, we include an illustrating example in appendix A and provide additional augment for the case $K = 2$ in appendix B.

**Shuffling and Splitpriors**     Algorithms 1, 2 and 3 describe the training process of DDFs. Note that only one denoising coupling layer is trained at a time. Moreover, we use invertible shuffle operations such that the input is partitioned differently in each coupling layer. Note that instead of propagating the full input vector $x$ through all layers of the DDF, we factor out parts of the input vector at regular intervals and model these parts conditioned on the other parts following the splitprior approach in (Dinh et al., 2016; Kingma & Dhariwal, 2018). As a result, the following coupling layers operate on lower-dimensional data. Not only is this more efficient, but it also allows for some additional dependencies between parts of the output vector $z$.

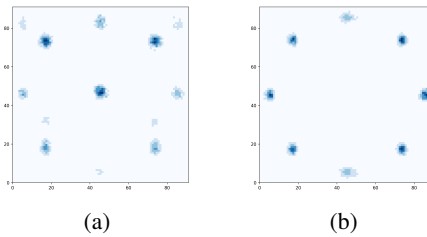

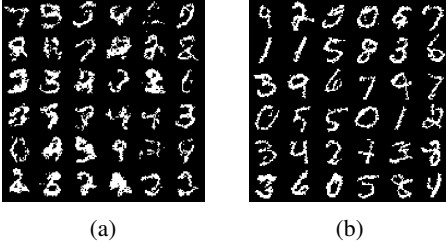

(a)              (b)                                    (a)              (b)

*Figure 2.* Qualitative results for (a) *Discrete Flows* and (b) *Discrete Denoising Flows* on the quantized eight Gaussians toy data set.

*Figure 3.* Qualitative results for (a) *Discrete Flows* and (b) *Discrete Denoising Flows* on the binarized MNIST data set.

## 4. Experiments

In this section we explore how the compression rate in bits per dimension (BPD) of *Discrete Denoising Flows* compares to *Discrete Flows* on three different data sets. Each experiment was conducted at least three times; we show the average results as well as the standard deviation in Table 1. To fully capture the difference in modeling capacity between DFs and DDFs, we trained both models with and without splitpriors. All experimental details and samples from the models trained without splitpriors can be found in appendices C and D. In each experiment, we use equally sized neural networks in the coupling layers of DFs and DDFs to ensure comparability.

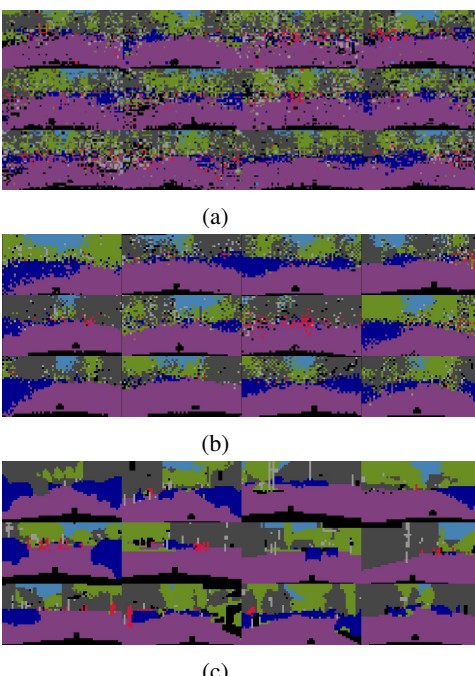

| DATA SET | SPLITPRIORS? | DF | DDF *(ours)* |
|---|---|---|---|
| 8 GAUSSIANS | NO | $5.05 \pm 0.05$ | $\mathbf{4.58} \pm 0.02$ |
| BIN. MNIST | NO | $0.37 \pm 0.01$ | $\mathbf{0.23} \pm 0.01$ |
| CITYSCAPES | NO | $1.46 \pm 0.00$ | $\mathbf{0.79} \pm 0.01$ |
| BIN. MNIST | YES | $0.17 \pm 0.01$ | $\mathbf{0.16} \pm 0.01$ |
| CITYSCAPES | YES | $0.65 \pm 0.03$ | $\mathbf{0.59} \pm 0.03$ |

*Table 1.* Comparison of achieved BPD of Discrete Flow (DF) and Denosing Discrete Flow (DDF) per data set.

**Eight Gaussians** As a first experiment, we train DDFs and DFs on a two-dimensional toy data set also used by Tran et al. (2019). This data set is a mixture of Gaussians with 8 means uniformly distributed around a circle and discretized into 91 bins (i.e. $K = 91$ classes). We model the data with a single coupling layer per model and set $h = K$ for the DDF coupling layer. For 2D no splitpriors are used, because that would already make the model universal and we cannot see how well the flow itself performs. As apparent from the qualitative results in Figure 2 as well as the achieved BPD given in Table 1, DDFs outperform DFs.

**Binary MNIST** In a second experiment, we train both DFs and DDFs on the binarized MNIST data set. Since the data set has $K = 2$ classes, we have $h = 2$ for the DDF coupling layers. The samples given in Figure 3 and the achieved BPDs in table 1 show that DDFs outperform DFs.

**Cityscapes** To test the performance of DDFs on image-type data, we use a 8-class version of the Cityscapes data set

*Figure 4.* Qualitative results for (a) *Discrete Flows* and (b) *Discrete Denoising Flows* on the Cityscapes data set. Figure (c) shows samples from the 8-class Cityscapes data set.

(Cordts et al., 2016) modified by Hoogeboom et al. (2021). This data set contains $32 \times 64$ segmentation maps, samples are given in figure 4c. Since we use multiple coupling layers in this experiment and have a data set with a number of classes $K > 2$, there is a trade-off between permuting classes and maintaining structure in each coupling layer. Therefore, after performing a grid search, we set $h = 4$ for all DDF coupling layers. From the samples in figure 4 and the BPD rates in table 1, we can see that DDFs outperform DFs.

## 5. Conclusion

In this paper, we have introduced a new discrete flow-based generative model for categorical data distributions, *Discrete Denoising Flows*. We showed that our model outperforms Discrete Flows in terms of log-likelihood.

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

## A. Denoising Coupling Layer: Example

Consider the distribution

$$P_{X_1 X_2}$$

| $x_1 \setminus x_2$ | 0 | 1 |
|---|---|---|
| 0 | 0.4 | 0.2 |
| 1 | 0.1 | 0.3 |

If we were to directly model $P_{X_1 X_2}$ with a factorized Bernoulli distribution

$$P_{model}(x) = \text{Bern}(x_1|p) \cdot \text{Bern}(x_2|q),$$

the highest achievable log likelihood would be

$$\mathbb{E}_{x \sim P_{X_1 X_2}}[\log_2 P_{model}(x)]$$
$$= \mathbb{E}_{x \sim P_{X_1 X_2}}[\log_2(\text{Bern}(x_1|p) \cdot \text{Bern}(x_2|q))]$$
$$\approx -1.97$$

with $p = 0.4$ and $q = 0.5$. Suppose now we have trained a neural net $n$ to predict $x_2$ from $x_1$, such that $\arg\max n(x_1) = x_2$. Using $n$ in a denoising coupling layer $T^{-1} : \mathcal{X} \to \mathcal{Z}$ as defined in equation 4 with $K = 2$ and implicitly $h = 2$ we obtain the distribution $P_{Z_1 Z_2}$. In this new distribution essentially the events with probability 0.3 and 0.1 are swapped.

$$P_{Z_1 Z_2}$$

| $z_1 \setminus z_2$ | 0 | 1 |
|---|---|---|
| 0 | 0.4 | 0.2 |
| 1 | 0.3 | 0.1 |

When modeling $P_{Z_1 Z_2}$ with a factorized Bernoulli distribution, the highest achievable log likelihood is

$$\mathbb{E}_{x \sim P_{X_1 X_2}}[\log_2 P_{model}(T^{-1}(x))]$$
$$= \mathbb{E}_{x \sim P_{Z_1 Z_2}}[\log_2(\text{Bern}(z_1|p) \cdot \text{Bern}(z_2|q))]$$
$$\approx -1.85$$

with $p = 0.4$ and $q = 0.7$. We can see that $P_{Z_1 Z_2}$ is now modeled higher log-likelihood than the factorized Bernoulli distribution on $P_{X_1 X_2}$.

## B. Denoising Coupling Layer: General Binary Case

In the following, we generalize the example given in appendix A to provide further insight into the functionality of the denoising coupling layer. Consider again a two-dimensional binary random variable $X$ with probability distribution $P_{X_1 X_2}$ defined as

$$P_{X_1 X_2}$$

| $x_1 \setminus x_2$ | 0 | 1 |
|---|---|---|
| 0 | $p_1$ | $p_2$ |
| 1 | $p_3$ | $p_4$ |

We train a neural network $n$ to predict $x_2$ from $x_1$, such that $\arg\max n(x_1) = x_2$. Using $n$ in a denoising coupling layer as defined in equation 4 with $K = 2$ and implicitly $h = 2$, we obtain the distribution $P_{Z_1 Z_2}$.

$$P_{Z_1 Z_2}$$

| $x_1 \setminus x_2$ | 0 | 1 |
|---|---|---|
| 0 | $\max(p_1, p_2)$ | $\min(p_1, p_2)$ |
| 1 | $\max(p_3, p_4)$ | $\min(p_3, p_4)$ |

We now want to show that applying the denoising coupling layer results in a distribution that can be modelled more accurately with a factorized Bernoulli distribution

$$P_{model}(x) = \text{Bern}(x_1|p) \cdot \text{Bern}(x_2|q),$$

with parameters $0 \leq p, q \leq 1$ in z-space than in x-space. To this end, we demonstrate that the model log-likelihood for $P_{Z_1 Z_2}$ is always higher than or equal to the model log-likelihood for $P_{X_1 X_2}$.

The model log-likelihood of $P_{X_1 X_2}$ is given as

$$\mathbb{E}_{x \sim P_{X_1 X_2}}[\log P_{model}(x)]$$
$$= \mathbb{E}_x[\log(\text{Bern}(x_1|p)) + \log(\text{Bern}(x_2|q))]$$
$$= \log(p) \cdot (p_1 + p_2) + \log(1 - p) \cdot (p_3 + p_4)$$
$$\quad + \log(q) \cdot (p_1 + p_3) + \log(1 - q) \cdot (p_2 + p_4)$$
$$= -\mathcal{H}(p) - \mathcal{H}(q)$$

where the last line assumes the optimal choice for $p = p_1 + p_2$ and $q = p_1 + p_3$ and $\mathcal{H}$ denotes the binary entropy defined as $\mathcal{H}(p) = -p \log p - (1 - p) \log(1 - p)$. Analogously for $P_{Z_1 Z_2}$ using Bernoulli parameters $\hat{q}, \hat{p}$

$$\mathbb{E}_{z \sim P_{Z_1 Z_2}}[\log P_{model}(z)]$$
$$= \log(\hat{p}) \cdot (\max(p_1, p_2) + \min(p_1, p_2))$$
$$\quad + \log(1 - \hat{p}) \cdot (\max(p_3, p_4) + \min(p_3, p_4))$$
$$\quad + \log(\hat{q}) \cdot (\max(p_1, p_2) + \max(p_3, p_4))$$
$$\quad + \log(1 - \hat{q}) \cdot (\min(p_1, p_2) + \min(p_3, p_4))$$
$$= -\mathcal{H}(\hat{p}) - \mathcal{H}(\hat{q})$$

with optimal choice $\hat{p} = \max(p_1, p_2) + \min(p_1, p_2) = p_1 + p_2$ which is the same, and the new $\hat{q} = \max(p_1, p_2) + \max(p_3, p_4)$. Since $\mathcal{H}(p) = \mathcal{H}(\hat{p})$ we only need to compare the terms containing $\hat{q}$ and $q$. We use that $-\mathcal{H}$ is a monotonic increasing function when only considering the

interval $[0.5, 1.0]$ and that $\mathcal{H}(p) = \mathcal{H}(1-p)$. First we check if $p_1 + p_3 \geq 0.5$ or otherwise $p_2 + p_4 \geq 0.5$. Let this value be $a$ so that $a \geq 0.5$, and the other value $b$ with $b \leq 0.5$. From the symmetry we have:

$$-\mathcal{H}(p_1 + p_3) = -\mathcal{H}(a) = -\mathcal{H}(p_2 + p_4) = -\mathcal{H}(b).$$

Next, observe that $\hat{q} = \max(p_1, p_2) + \max(p_3, p_4) \geq \max(p_1 + p_3, p_2 + p_4) = a$ and since $-\mathcal{H}$ is monotonically increasing on $[0.5, 1.0]$ it follows that:

$$-\mathcal{H}(\hat{q}) \geq -\mathcal{H}(a) = -\mathcal{H}(q).$$

Plugging this back into the previous equation gives us the desired identity:

$$\mathbb{E}_{z \sim P_{Z_1 Z_2}}[\log P_{model}(z)] \geq \mathbb{E}_{x \sim P_{X_1 X_2}}[\log P_{model}(x)]$$

under optimal choice of the Bernoulli parameters.

## C. Experimental Details

We train *Discrete Flows* and *Discrete Denoising Flows* on three data sets. In each experiment, both models use the same architecture to ensure comparability.

Throughout the experiments, we use the Adam optimizer, a learning rate of $0.001$, and a batch size of $64$. The base distribution is always a factorized categorical distribution with $K$ classes. $K$ varies between the data sets. Recall that in the coupling layers of both DFs and DDFs, the $D$-dimensional coupling layer input $x$ is split into two parts such that $x = [x_a, x_b]$, at a split index $d$. For all of our experiments, we set $d = \frac{D}{2}$.

**Eight Gaussians**  In this experiment, we train a single-layer *Discrete Flow* and a single-layer *Discrete Denoising Flow*. For both models, we use an MLP consisting of $4$ linear layers with 256 hidden units and ReLU activations to parameterize the coupling layer. This small model size is sufficient for modeling our 2D toy data set.

Consisting of only one coupling layer, preserving the structure of the input vector for later coupling layers is not relevant for the DDF model. Consequently, we set the parameter $h$ in the denoising coupling layer to the number of classes in the data set $K = 91$.

**Binary MNIST**  In this experiment, we work with binary image data. Consequently, each DF and DDF coupling layer is parameterized by a DenseNet (Huang et al., 2017) consisting of $8$ dense building blocks. For DDFs, modeling binary data implies that $h$ equals the number of classes $K$, i.e. $h = K = 2$.

We embed the coupling layers into a multi-layer architecture of coupling layers, split priors, and *squeeze* operations (Dinh et al., 2016). The *squeeze* changes the vector size from $[channels \times H \times W]$ to $[4 \cdot channels \times \frac{H}{2} \times \frac{W}{2}]$.

The overall model architecture consists of 2 blocks that consisting of the following layers (in that order):

{*squeeze - coupling - splitprior - coupling - splitprior*}

Note that each coupling layer is preceded by a shuffling operation applied to the channels of the input vector. Further the splitprior factors out the opposite part that the coupling transformed (so if the coupling layer transforms $x_b \mapsto z_b$ then $z_a$ is factored out).

**Cityscapes**  In this experiment, we're again dealing with image-type data, this time with $K = 8$ classes. Like in the previous experiment, we utilize a DenseNet (Huang et al., 2017) in the DF and DDF coupling layers. However, here it consists of $15$ dense building blocks. We perform a grid search for the DDF parameter $h$ on $\{1, 2, 4, 6, 8\}$ and find that $h = 4$ leads to the best performance. Since the last coupling layer does not have a trade-off between permuting classes and maintaining structure for the next coupling layer to work on, we can set $h = K = 8$ in the last layer.

In analogy to the previous experiment, we embed the coupling layers in a multi-layer architecture of coupling layers, splitpriors and squeeze operations. For this experiment, the model architecture consists of 3 building blocks with the following layers (in that order):

{*squeeze - coupling - splitprior - coupling - splitprior*}

Again, the splitprior factors out the opposite part that the coupling transformed (so if the coupling layer transforms $x_b \mapsto z_b$ then $z_a$ is factored out).

## D. Training without splitpriors

In the experiments without splitpriors, we propagate the full input vector $x$ through all layers of the *Discrete Flow* and *Discrete Denoising Flow* models. We perform this additional set of experiments to further illustrate the difference in modeling capacity between *Discrete Flows* and *Discrete Denoising Flows*. When both models are trained with splitpriors, the splitpriors allow for additional dependencies between the input and output of the model, thus increasing modeling capacity. If we omit the splitpriors, the model performance depends solely on the coupling layers, allowing us to better see the difference in performance between the two types of flows.

### D.1. Experimental Setup

Similar to the experiments with splitpriors, we use the Adam optimizer, a learning rate of $0.001$, a batch size of $64$ and set the split index $d = \frac{D}{2}$ with $D$ being the dimensionality of the input $x$ for all experiments without splitpriors.

**Binary MNIST** Equivalent to the MNIST binary experiment described in Appendix C, each DF and DDF coupling layer is parameterized by a DenseNet (Huang et al., 2017) consisting of 8 dense building blocks, and we set $h = K = 2$ for the DDF model. The overall model architecture for this experiment consists of 2 building blocks with the following layers (in that order):

$$\{squeeze \text{ - } coupling \text{ - } coupling \}$$

where each coupling layer is preceded by a shuffling operation applied to the channels of the input vector.

**Cityscapes** Like in the Cityscapes experiment described in Appendix C, each DF and DDF coupling layer is parameterized by a DenseNet (Huang et al., 2017) consisting of 15 dense building blocks, and we set $h = 4$ for each DDF coupling layer but the last, where we set $h = K = 8$.

The model architecture for this experiment consists of 1 building block with layers (in that order)

$$\{squeeze \text{ - } coupling \}$$

followed by 2 building blocks with layers (in that order)

$$\{squeeze \text{ - } coupling \text{ - } coupling \}$$

where each coupling layer is preceded by a shuffling operation applied to the channels of the input vector.

### D.2. Evaluation

Figure 5 and figure 6 show samples from the trained DF and DDF models on the binary MNIST data set and on the Cityscapes data set. As we can see from the samples as well as from the quantitative results given in table 1, there is a bigger difference in performance between DFs and DDFs when they are trained without splitpriors. The reason is that splitpriors already increase the model capacity, making the relative difference smaller. Therefore, this comparison shows even more clearly the improved modeling performance of our DDFs over DFs.

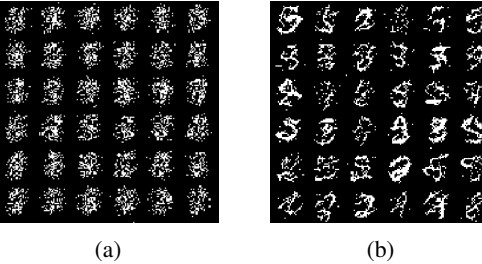

(a)                                     (b)

*Figure 5.* Qualitative results for (a) *Discrete Flows* and (b) *Discrete Denoising Flows* trained without splitpriors on the binarized MNIST data set.

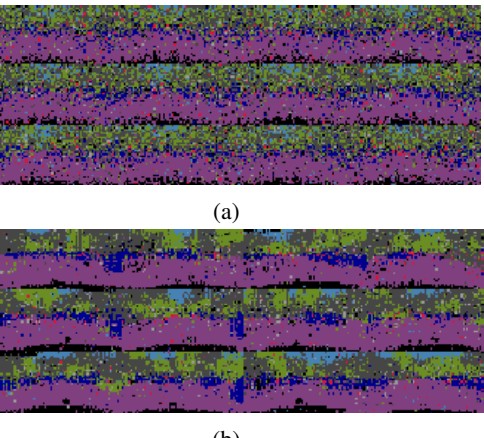

(a)

(b)

*Figure 6.* Qualitative results for (a) *Discrete Flows* and (b) *Discrete Denoising Flows* trained without splitpriors on the Cityscapes data set