# OpenReview forum: "Discrete Denoising Flows"
_ICML.cc/2021/Workshop/INNF — INNF+ 2021 spotlighttalk_

### Official Review · Reviewer_wYzL · 2021-06-10

**Rating:** Accept
**Confidence:** 4

**Summary:**

**Summary of paper**

Motivated by the lossless compression capabilities of discrete flows, this paper introduces a novel coupling layer operator based on the argsort operation.  They propose a greedy algorithm that uses classifiers at each stage for training. They demonstrate their approach is better than discrete flows for three datasets.

**Summary of review**

Overall, interesting new coupling layer for discrete flows and simple greedy training algorithm.  Solid fit for the workshop and for continued discussion.


**Justification For Rating:**

**Strengths:**
- Introduce new coupling layer for discrete flows based on the argsort operator.
- Propose novel training meta-algorithm that only trains a simple classifier at each stage.
- Demonstrated better performance on several cases over bipartite discrete flows from Tran et al. 2019.

**Weaknesses:**
- Theoretical grounding for training approach seems missing.  Only some intuition is given but is fairly confusing.  I think the intuition is right but formalizing this would definitely help the paper.

**Other comments or questions**
- Algorithm is greedy.  Though I am a fan of greedy algorithms for their simplicity.  Is there a reason this cannot be done using global training?  It seems if you were able to approximate argsort (similar to approximating argmax with softmax), then you could do joint training.

- For the $argsort\_top_h$ function, it looks like this is a greedy pairwise swap to get the top h. But mathematically, it doesn't seem that there has to be a unique solution to this operation (unless you assume some sequential algorithm to compute this operation). Suppose $\theta = [10,20,30,40]$, then two permutations for h=2, could be [4,3,2,1] or [4,3,1,2].

- I may suggest avoiding the title "denoising" since this term suggests something like denoising autoencoders or denoising diffusiion probabilistic models, i.e., where noise is injected into the training process somehow.

---

### Official Review · Reviewer_ybK3 · 2021-06-11

**Rating:** Accept
**Confidence:** 3

**Summary:**

The paper introduces a new approach to NFs on discrete data. The key is a denoising decoupling layer which explicitly learns a (partial) permutation of the input that is conditioned on the output of a neural network. Evaluation shows that the model outperforms discrete flows on three baselines (one synthetic, to real-world) when evaluated on compression (bits-per-dimension).

**Justification For Rating:**

The work is novel, and the write-up is precise and easy to follow. The evaluation is sufficiently convincing for the venue.

I would have preferred a little more discussion on downsides and limitations. Unless I'm mistaken, it seems that the model requires two-pass training, to learn the classifiers in Algorithm 1. How many of these are required, and how does this impact training time and memory use (especially when compared to DFs)? The h parameter appears to offer a tradeoff between performance and complexity. Some discussion/investigation of this tradeoff would be interesting (or, if the impact is different, then a discussion of how we shoudl interpret h).

PixelCNN is mentioned as related work but not evaluated as a baseline. While I appreciate that PixelCNN is very expensive to run, training a DDF on CIFAR10 or Imagenet64 should allow for a comparison against the published BPD of PixelCNNs.

Pet peeve: I would suggest using the word "use" rather than "utilize".

---

### Official Review · Reviewer_6y2X · 2021-06-12

**Rating:** Accept
**Confidence:** 5

**Summary:**

This paper contributes toward the efforts of devising expressive flow models for discrete data, which differ in their practical requirements because tractable calculation of the determinant of the jacobian is no longer needed. Instead, a main issue was backprop through a quantization step that relied on biased gradients.

The authors propose an alternative to how discrete flows have previously  been approached by introducing a coupling layer based on conditional permutation of the categorical variable. They outline its invertibility and mechanisms for training them and then provide preliminary experiments benchmarking them against Discrete Flows (Tran et al 2019) on a few tasks: a toy Gaussian, MNIST, and Cityscapes. Their Denoisinig Discrete Flows perform better across the metric of compression rate in bits per dimensions.

**Justification For Rating:**

This paper seems like a useful contribution to the workshop. It makes a clear, “discrete” (pun intended) contribution to the efforts of making discrete flows useful and has the b eginnings of experimental support for its efficacy.

The reviewer has a few remarks:

- The authors make a point that the state-of-the-art in compression, PixelCNN, has costly decoding. However, the model they introduce in Discrete Denoising flows seems to involve its own costly practices in that it must be trained layer by layer. Do the authors see a way of extending this to end-to-end training to avoid this pitfall?
- It would be nice to see benchmarks of DDF against the state-of-the-art lossless compression, namely the PixelCNN above.


As this is a workshop paper, this should be included in a longer form of the work in the future.

---

### Decision · Program_Chairs · 2021-06-14

Accept (spotlight talk)